# GENERALIZING GRAPH CONVOLUTIONAL NETWORKS VIA HEAT KERNEL

## ABSTRACT

Graph convolutional networks (GCNs) have emerged as a powerful framework for mining and learning with graphs. A recent study shows that GCNs can be simplified as a linear model by removing nonlinearities and weight matrices across all consecutive layers, resulting the simple graph convolution (SGC) model. In this paper, we aim to understand GCNs and generalize SGC as a linear model via heat kernel (HKGCN), which acts as a low-pass filter on graphs and enables the aggregation of information from extremely large receptive fields. We theoretically show that HKGCN is in nature a continuous propagation model and GCNs without nonlinearities (i.e., SGC) are the discrete versions of it. Its low-pass filter and continuity properties facilitate the fast and smooth convergence of feature propagation. Experiments on million-scale networks show that the linear HKGCN model not only achieves consistently better results than SGC but also can match or even beat advanced GCN models, while maintaining SGC's superiority in efficiency.

## 1 INTRODUCTION

Graph neural networks (GNNs) have emerged as a powerful framework for modeling structured and relational data (Gori et al., 2005; Scarselli et al., 2008; Gilmer et al., 2017; Kipf & Welling, 2017). A wide range of graph mining tasks and applications have benefited from its recent emergence, such as node classification (Kipf & Welling, 2017; Veličković et al., 2018), link inference (Zhang & Chen, 2018; Ying et al., 2018), and graph classification (Xu et al., 2019b).

The core procedure of GNNs is the (discrete) feature propagation operation, which propagates information between nodes layer by layer based on rules derived from the graph structures. Take the graph convolutional network (GCN) (Kipf & Welling, 2017) for example, its propagation is performed through the normalized Laplacian of the input graph. Such a procedure usually involves 1) the non-linear feature transformation, commonly operated by the activation function such as ReLU, and 2) the discrete propagation layer by layer. Over the course of its development, various efforts have been devoted to advancing the propagation based architecture, such as incorporating self-attention in GAT (Veličković et al., 2018), mixing high-order neighborhoods in MixHop (Abu-El-Haija et al., 2019), and leveraging graphical models in GMNN (Qu et al., 2019).

Recently, Wu et al. (Wu et al., 2019) observe that the non-linear part of GCNs' feature propagation is actually associated with excess complexity and redundant operations. To that end, they simplify GCNs into a linear model SGC by removing all non-linearities between consecutive GCN layers. Surprisingly, SGC offers comparable or even better performance to advanced GCN models, based on which they argue that instead of the non-linear feature transformation, the repeated graph propagation may contribute the most to the expressive power of GCNs.

Though interesting results generated, SGC still inherits the discrete nature of GCNs' propagation, which can lead to strong oscillations during the procedure. Take, for example, a simple graph of two nodes $v_1$ and $v_2$ with one-dimension input features $\mathbf{x}_1 = 1$ & $\mathbf{x}_2 = 2$ and one weighted edge between them, the feature updates of $x_1$ and $x_2$ during the GCN propagation is shown in Figure 1 (a), from which we can clearly observe the oscillations of $x_1$ and $x_2$ step by step. This indicates that though the features from multi-hops away may seem to be taken into consideration during the GCN propagation, it is still far away to learn patterns from them.

In this work, we aim to generalize GCNs into a continuous and linear propagation model, which is referred to as HKGCN. We derive inspiration from Newton's law of cooling by assuming graph feature propagation follow a similar process. Straightforwardly, this leads us to leverage heat kernel for feature propagation in HKGCN. Theoretically, we show that the propagation matrix of GCNs is equivalent to the finite difference version of the heat kernel. In other words, using heat kernel as the propagation matrix will lead to smooth feature convergence. In the same example above, we show the heat kernel based propagation in HKGCN can prevent oscillations, as illustrated in Figure 1 (b). Finally, from the graph spectral perspective, heat kernel acts as a low-pass filter and the cutoff frequency of heat kernel can be adjusted by changing the propagation time.

Empirically, we demonstrate the performance of HKGCN for both transductive and inductive semi-supervised node classification tasks. The experiments are conducted on both traditional GNN datasets, such as Cora, CiteSeer, Pubmed, and Reddit, and latest graph benchmark data indexed by Open Graph Benchmark (Hu et al., 2020). The results suggest that the simple and linear HKGCN model can consistently outperform SGC on all six datasets and match or even beat the performance of advanced graph neural networks on both tasks, while at the same time maintaining the order-of-magnitude efficiency superiority inherited from SGC.

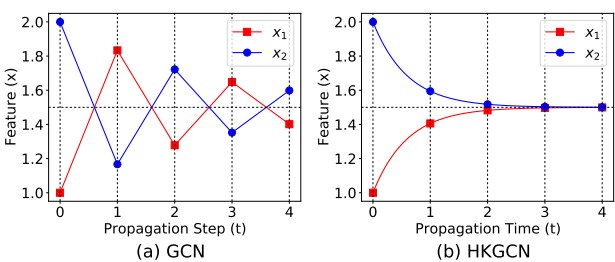

Figure 1: Feature propagation under GCN and HKGCN.

## 2 RELATED WORK

**Graph Neural Networks.** Graph neural networks (GNNs) have emerged as a new paradigm for graph mining and learning, as significant progresses have been made in recent years. Notably, the spectral graph convolutional network (Bruna et al., 2013) is among the first to directly use back propagation to learn the kernel filter, but this has the shortcoming of high time complexity. Another work shows how to use Chebyshev polynomial approximation to fast compute the filter kernel (Hammond et al., 2011). Attempts to further this direction leverage Chebyshev Expansion to achieve the same linear computational complexity as classical CNNs (Defferrard et al., 2016). Later, the graph convolutional network (GCN) (Kipf & Welling, 2017) simplifies the filter kernel to the second-order of Chebyshev Expansion, inspiring various advancements in GNNs. GAT brings the attention mechanisms into graph neural networks (Veličković et al., 2018). GMNN combines the benefits of statistical relational learning and GNNs into a unified framework (Qu et al., 2019). To enable fast and scalable GNN training, FastGCN interprets graph convolutions as integral transforms of features and thus uses Monte Carlo method to simulate the feature propagation step (Chen et al., 2018). GraphSage treats the feature propagation as the aggregation from (sampled) neighborhoods (Hamilton et al., 2017). LADIES (Zou et al., 2019) further introduces the layer-dependent importance sampling technique for efficient training.

Recently, there are also research efforts devoting on the theoretical or deep understanding of GCNs (Xu et al., 2019b; Battaglia et al., 2018). For example, the feature propagation in GNNs can be also explained as neural message passing (Gilmer et al., 2017). In addition, studies also find that the performance of GNNs decreases with more and more layers, known as the over-smoothing issue (Li et al., 2018; Zhao & Akoglu, 2020). To reduce GCNs' complexity, SGC turns the GCN model into a linear model by removing the non-linear activation operations between consecutive GCN layers (Wu et al., 2019), producing promising results in terms of both efficacy and efficiency.

**Heat Kernel.** The properties of heat kernel for graphs are reviewed in detail by Chuang in (Chung & Graham, 1997). Recently, heat kernel has been frequently used as the feature propagation modulator. In (Kondor & Lafferty, 2002), the authors show that heat kernel can be regarded as the discretization of the familiar Gaussian kernel of Euclidean space. Additionally, heat kernel is often used as the window function for windowed graph Fourier transform (Shuman et al., 2016). In (Zhang et al., 2019), the second-order heat kernel is used as the band-pass filter kernel to amplify local and global structural information for network representation learning.

**Concurrent work.** Several recent works have developed similar idea. (Poli et al., 2020; Zhuang et al., 2020) use the Neural ODE framework and parametrize the derivative function using a 2 or 3 layer GNN directly. (Xhonneux et al., 2020) improved ODE by developing a continuous message-passing layer. All ODE models make feature converge to stable point by adding residual connection. In contrast, our model outputs an intermediate state of feature, which is a balance between local and global features. Some recent works (Xu et al., 2019a; Klicpera et al., 2019) propose to leverage heat kernel to enhance low-frequency filters and enforce a smooth feature propagation. However, they do not realize the relationship between the feature propagation of GCNs and heat kernel.

## 3 GENERALIZING (SIMPLE) GRAPH CONVOLUTION VIA HEAT KERNEL

### 3.1 PROBLEM AND BACKGROUND

We focus on the problem of semi-supervised node classification on graphs, which is the same as GCN (Kipf & Welling, 2017). Without loss of generality, the input to this problem is an undirected network $G = (V, E)$, where $V$ denotes the node set of $n$ nodes $\{v_1, ..., v_n\}$ and $E$ represents the edge set. The symmetric adjacency matrix of $G$ is defined as $\mathbf{A}$ and its diagonal degree matrix as $\mathbf{D}$ with $\mathbf{D}_{ii} = \sum_j \mathbf{A}_{ij}$. For each node $v_i \in V$, it is associated with a feature vector $\mathbf{x_i} \in \mathbf{X} \in \mathbb{R}^{n \times d}$ and a one-hot label vector $\mathbf{y_i} \in \mathbf{Y} \in \{0, 1\}^{n \times C}$, where $C$ is the number of classes. The problem setting of semi-supervised graph learning is given the labels $\mathbf{Y}_L$ of a subset of nodes $V_L$, to infer the labels $\mathbf{Y}_U$ of the remaining nodes $V_U$, where $V_U = V \backslash V_L$.

**Graph Convolutional Networks.** Given the input graph $G = (V, E)$ with $\mathbf{A}$, $\mathbf{D}$, $\mathbf{X}$, and $\mathbf{Y_L}$, GCN can be understood as feature propagation over the graph structure. Specifically, it follows the following propagation rule:

$$\mathbf{H}^{(l+1)} = \sigma(\tilde{\mathbf{D}}^{-\frac{1}{2}}\tilde{\mathbf{A}}\tilde{\mathbf{D}}^{-\frac{1}{2}}\mathbf{H}^{(l)}\mathbf{W}^{(l)}), \tag{1}$$

where $\tilde{\mathbf{A}} = \mathbf{A} + \mathbf{I}_N$ is the adjacency matrix with additional self-connections with $I_N$ as the identity matrix, $\mathbf{W}^{(l)}$ is a trainable weight matrix in the $l^{th}$ layer, $\sigma(\cdot)$ is a nonlinear function such as ReLU, and $\mathbf{H}^{(l)}$ denotes the hidden node representation in the $l^{th}$ layer with the first layer $\mathbf{H}^{(0)} = \mathbf{X}$.

The essence of GCN is that each GCN layer is equivalent to the first-order Chebyshev expansion of spectral convolution (Kipf & Welling, 2017). It also assumes that the first-order coefficient $a_1$ is equal to the 0-th order coefficient $a_0$ multiplied by $-1$, i.e., $a_1 = -a_0$. We will later prove that this is just a discrete solution of heat equation.

**Simple Graph Convolution.** Since its inception, GCNs have drawn tremendous attention from researchers (Chen et al., 2018; Veličković et al., 2018; Qu et al., 2019). A recent study shows that GCNs can be simplified as the Simple Graph Convolution (SGC) model by simply removing the nonlinearities between GCN layers (Wu et al., 2019). Specifically, the SGC model is a linear model and can be formalized by the following propagation rule:

$$\mathbf{Y} = softmax((\tilde{\mathbf{D}}^{-\frac{1}{2}}\tilde{\mathbf{A}}\tilde{\mathbf{D}}^{-\frac{1}{2}})^K \mathbf{X}\mathbf{W}) \tag{2}$$

Surprisingly, the linear SGC model yields comparable prediction accuracy to the sophisticated GCN models in various downstream tasks, with significant advantages in efficiency and scalability due to its simplicity.

**Heat Equation and Heat Kernel.** The heat equation, as a special case of the diffusion equation, is used to describe how heat distributes and flows over time (Widder & Vernon, 1976).

Image a scenario of graph, in which each node has a temperature and heat energy could only transfer along the edge between connected nodes, and the heat propagation on this graph follows Newton's law of cooling. So the heat propagation between node $v_i$ and node $v_j$ should be proportional to 1) the edge weight and 2) the temperature difference between $v_i$ and $v_j$. Let $\mathbf{x}_i^{(t)}$ denote the temperature of $v_i$ at time $t$, the heat diffusion on graph $G$ can be described by the following heat equation:

$$\frac{d\mathbf{x}_i^{(t)}}{dt} = -k \sum_j \mathbf{A}_{ij}(\mathbf{x}_i^{(t)} - \mathbf{x}_j^{(t)}) = -k[\mathbf{D}_{ii}\mathbf{x}_i^{(t)} - \sum_j \mathbf{A}_{ij}\mathbf{x}_j^{(t)}]. \tag{3}$$

The equation under the matrix form is $\frac{d\mathbf{X}^{(t)}}{dt} = -k\mathbf{L}\mathbf{X}^{(t)}$, where $\mathbf{L} = \mathbf{D} - \mathbf{A}$ is the graph Laplacian matrix. By reparameterizing $t$ and $k$ into a single term $t' = kt$, the equation can be rewritten as:

$$\frac{d\mathbf{X}^{(t')}}{dt'} = -\mathbf{L}\mathbf{X}^{(t')} \tag{4}$$

A heat kernel is the fundamental solution of the heat equation (Chung & Graham, 1997). The heat kernel $H_t$ is defined to be the $n \times n$ matrix:

$$\mathbf{H}_t = e^{-\mathbf{L}t} \tag{5}$$

Given the initial status $\mathbf{X}^{(0)} = \mathbf{X}$, the solution to the heat equation in Eq. 4 can be written as

$$\mathbf{X}^{(t)} = \mathbf{H}_t\mathbf{X} \tag{6}$$

Naturally, the heat kernel can be used as the feature propagation matrix in GCNs.

## 3.2 Connecting GCN and SGC to Heat Kernel

GCN's feature propagation follows $\tilde{\mathbf{D}}^{-\frac{1}{2}}\tilde{\mathbf{A}}\tilde{\mathbf{D}}^{-\frac{1}{2}}$, through which node features diffuse over graphs. Note that the feature propagation in GCN is just one step each time/layer, hindering individual nodes from learning global information. By analogy with the heat diffusion process on graphs, the heat kernel solution can also be perfectly generalized to the feature propagation in the graph convolution.

Instead of using $\mathbf{L}$, we follow GCN to use the symmetric normalized Laplacian $\tilde{\mathbf{L}} = \mathbf{I} - \tilde{\mathbf{D}}^{-\frac{1}{2}}\tilde{\mathbf{A}}\tilde{\mathbf{D}}^{-\frac{1}{2}}$ to replace it. According to SGC (Wu et al., 2019), this convert also serves as a low-pass-type filter in graph spectral. Then we have Eq. 4 as $\frac{d\mathbf{X}^{(t)}}{dt} = -\tilde{\mathbf{L}}\mathbf{X}^{(t)}$ and the heat kernel in Eq. 5 as $\mathbf{H}_t = e^{-\tilde{\mathbf{L}}t}$. Consider the finite difference of this heat equation, it could be written as:

$$\frac{\mathbf{X}^{(t+\Delta t)} - \mathbf{X}^{(t)}}{\Delta t} = -\tilde{\mathbf{L}}\mathbf{X}^{(t)} \tag{7}$$

If we set $\Delta t = 1$, we have $\mathbf{X}^{(t+1)} = \mathbf{X}^{(t)} - \tilde{\mathbf{L}}\mathbf{X}^{(t)} = \tilde{\mathbf{D}}^{-\frac{1}{2}}\tilde{\mathbf{A}}\tilde{\mathbf{D}}^{-\frac{1}{2}}\mathbf{X}^{(t)}$. This is the same feature propagation rule in Eq. 1. In other words, each layer of GCN's feature propagation on graphs is equal to the finite difference of the heat kernel. If we consider the multilayer GCN without the activation function, it could be written as SGC (Wu et al., 2019): $\mathbf{Y} = softmax((\tilde{\mathbf{D}}^{-\frac{1}{2}}\tilde{\mathbf{A}}\tilde{\mathbf{D}}^{-\frac{1}{2}})^K\mathbf{X}\mathbf{W})$, where $\mathbf{Y}$ is the classification result and $\mathbf{W}$ is the merged weight matrix. Using the finite difference of heat equation above, it can be rewritten as:

$$\mathbf{Y} = softmax(\mathbf{X}^{(K)}\mathbf{W}), \tag{8}$$

which is still a multistep finite difference approximation of the heat kernel.

**Reduce $\Delta t$.** Assuming $t$ can be divided by $\Delta t$, the number of iterations is $n_i = \frac{t}{\Delta t}$, making Eq. 7 become $\mathbf{X}^{(t)} = (\mathbf{I} - \Delta t\tilde{\mathbf{L}})^{n_i}\mathbf{X}$. By fixing $t = 1$, we have

$$\begin{aligned}
\mathbf{X}^{(1)} &= (\mathbf{I} - \frac{1}{n_i}\tilde{\mathbf{L}})^{n_i}\mathbf{X} \\
&= (\mathbf{I} - \frac{n_i}{n_i}\tilde{\mathbf{L}} + \frac{n_i(n_i-1)}{n_i^2}\frac{\tilde{\mathbf{L}}^2}{2!} + \cdots + \frac{(-1)^{n_i}n_i!}{n_i^{n_i}}\frac{\tilde{\mathbf{L}}^{n_i}}{n_i!})\mathbf{X}
\end{aligned} \tag{9}$$

Therefore, $n_i$ iterations of $t = 1$ approximate to Taylor expansion of the heat kernel at order $n_i$. So GCN could also be seen as the first-order Taylor expansion of the heat kernel.

## 3.3 Heat Kernel as Feature Propagation

We have shown that the feature propagation in GCN is a multistep finite difference approximation of the heat kernel. Next, we briefly illustrate the advantage of differentiation.

**A case study.** We illustrate how the features are updated during the GCN and heat kernel propagations. Let us consider a graph of two nodes $v_1$ and $v_2$ with one-dimension input features $\mathbf{x}_1 = 1$ and $\mathbf{x}_2 = 2$ and one weighted edge $\mathbf{A}_{12} = \mathbf{A}_{21} = 5$ between them. Recall that with $\Delta t = 1$ GCN is equivalent to the finite difference of the heat kernel. Thus, we set $\Delta t = 1$ for the heat kernel as well.

The updates of $\mathbf{x}_1$ and $\mathbf{x}_2$ as the propagation step $t$ increases are shown in Figure 1. We can observe that heat kernel shows much smoother and faster convergence than GCN.

In GCN, the discrete propagation $\tilde{\mathbf{D}}^{-\frac{1}{2}}\tilde{\mathbf{A}}\tilde{\mathbf{D}}^{-\frac{1}{2}}$ layer by layer causes node features to keep oscillating around the convergence point. The reason lies in GCN's requirement for $\Delta t = 1$, which is too large to have a smooth convergence. Straightforwardly, the oscillating nature of GCN's feature propagation makes it sensitive to hyper-parameters and generate weak performance on large graphs. Theoretical analysis is given in Section 3.5.

## 3.4 GENERALIZING GRAPH CONVOLUTION

We have shown that the feature propagation in GCN is merely the solution of the finite difference version of the heat equation when $\Delta t = 1$. As a result, using heat kernel as feature propagation can lead to smooth convergence. Since the range of $t$ is in real number field, the propagation time $t$ in heat kernel can be seen as a generalized parameter of the number of layers in GCN (Kipf & Welling, 2017) and SGC (Wu et al., 2019). The advantage of heat kernel also includes that $t$ can change smoothly compared to the discrete parameters.

In light of this, we propose to generalize graph convolution networks by using heat kernel and present the HKGCN model. Specifically, we simply use the one layer linear model:

$$\mathbf{Y} = softmax(\mathbf{X}^{(t)}\mathbf{W}) = softmax(e^{(-\tilde{\mathbf{L}}t)}\mathbf{X}\mathbf{W}), \tag{10}$$

where $\mathbf{W}$ is the $n \times C$ feature transformation weight and $t$ can be a learnable scalar or a preset hyper-parameter.

Using a preset $t$ converts HKGCN to 1) a pre-processing step $\bar{\mathbf{X}} = e^{(-\tilde{\mathbf{L}}t)}\mathbf{X}$ without parameters and 2) a linear logistic regression classifier $\mathbf{Y} = softmax(\bar{\mathbf{X}}\mathbf{W})$. This makes the training speed much faster than GCN. The algorithm of presetting $t$ is in Appendix (Algorithm 1).

To avoid eigendecomposition, we use Chebyshev expansion to calculate $e^{(-\tilde{\mathbf{L}}t)}$ (Hammond et al., 2011; Zhang et al., 2019). The first kind Chebyshev polynomials are defined as $T_{i+1}(x) = 2xT_i(x) - T_{i-1}(x)$ with $T_0(x) = 1$ and $T_1(x) = x$. The requirement of the Chebyshev polynomial is that $x$ should be in the range of $[-1, 1]$, however, the eigenvalues of $\tilde{\mathbf{L}}$ satisfy $0 = \lambda_0 \leq ... \leq \lambda_{n-1} \leq 2$. To make the eigenvalues of $\bar{\mathbf{L}}$ fall in the range of $[-1, 1]$, we convert $\tilde{\mathbf{L}}$ to $\bar{\mathbf{L}} = \tilde{\mathbf{L}}/2$. In addition, we reparameterize $t$ to $\tilde{t} = 2t$ so that the heat kernel keeps in the original form. In doing so, we have

$$e^{(-\tilde{\mathbf{L}}t)} = e^{(-\bar{\mathbf{L}}\tilde{t})} \approx \sum_{i=0}^{k-1} c_i(\tilde{t})T_i(\bar{\mathbf{L}}) \tag{11}$$

And the coefficient $c_i$ of the Chebyshev expansion can be obtained by:

$$c_i(\tilde{t}) = \frac{\beta}{\pi}\int_{-1}^{1}\frac{T_i(x)e^{-x\tilde{t}}}{\sqrt{1-x^2}}dx = \beta(-1)^i B_i(\tilde{t}) \tag{12}$$

where $\beta = 1$ when $i = 0$, otherwise $\beta = 2$, and $B_i(\tilde{t})$ is the modified Bessel function of the first kind (Andrews & of Photo-optical Instrumentation Engineers, 1998). By combining Eqs. 11 and 12 together, we can have $e^{(-\bar{\mathbf{L}}\tilde{t})}$ approximated as:

$$e^{(-\bar{\mathbf{L}}\tilde{t})} \approx B_0(\tilde{t})T_0(\bar{\mathbf{L}}) + 2\sum_{i=1}^{k-1}(-1)^i B_i(\tilde{t})T_i(\bar{\mathbf{L}}) \tag{13}$$

## 3.5 SPECTRAL ANALYSIS

From the graph spectral perspective, heat kernel acts as a low-pass filter (Xu et al., 2019a). In addition, as the propagation time $t$ increases, the cutoff frequency decreases, smoothing the feature propagation.

**Graph Spectral Review.** We define $\mathbf{\Lambda} = diag(\lambda_1, ..., \lambda_n)$ as the diagonal matrix of eigenvalues of $\tilde{\mathbf{L}}$ and $\mathbf{U} = (\mathbf{u}_1, ..., \mathbf{u}_n)$ as the corresponding eigenvectors, that is, $\tilde{\mathbf{L}} = \mathbf{U}\mathbf{\Lambda}\mathbf{U}^T$.

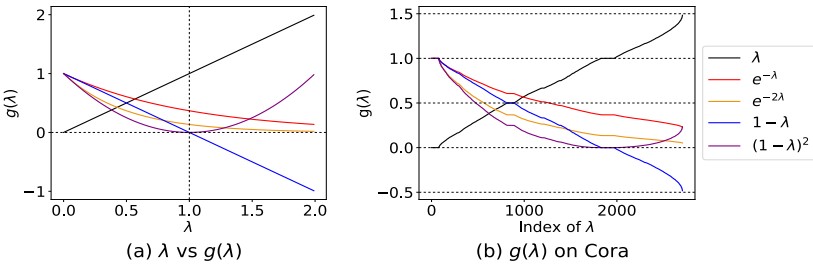

Figure 2: Effects of different kernels.

The graph Fourier transform defines that $\hat{\mathbf{x}} = \mathbf{U}^T \mathbf{x}$ is the frequency spectrum of $\mathbf{x}$, with the inverse operation $\mathbf{x} = \mathbf{U}\hat{\mathbf{x}}$. And the graph convolution between the kernel filter $g(\cdot)$ and $\mathbf{x}$ is $\mathbf{g} * \mathbf{x} = \mathbf{U}g(\mathbf{\Lambda})\mathbf{U}^T \mathbf{x}$ with $g(\mathbf{\Lambda}) = diag(g(\lambda_1), \cdots, g(\lambda_n))$.

For a polynomial $g(\tilde{\mathbf{L}})$, we have $g(\tilde{\mathbf{L}}) = \mathbf{U}g(\mathbf{\Lambda})\mathbf{U}^T$. This can be verified by setting $g(\lambda_i) = \sum_{j=0}^{K} a_j \lambda_i^j$, that is, $\mathbf{U}g(\mathbf{\Lambda})\mathbf{U}^T = \mathbf{U}\sum_{j=0}^{K} a_j \mathbf{\Lambda}^j \mathbf{U}^T = \sum_{j=0}^{K} a_j \mathbf{U}\mathbf{\Lambda}^j \mathbf{U}^T = \sum_{j=0}^{K} a_j \tilde{\mathbf{L}}^j = g(\tilde{\mathbf{L}})$.

**Heat Kernel.** As $g(\tilde{\mathbf{L}}) = \mathbf{U}g(\mathbf{\Lambda})\mathbf{U}^T$, the heat kernel $\mathbf{H}_t = e^{-\tilde{\mathbf{L}}t} = \sum_{i=0}^{\infty} \frac{1}{i!}t^i \tilde{\mathbf{L}}^i$ could also be seen as a polynomial of $\tilde{L}$. Thus its kernel filter is $g(\lambda_i) = e^{-\lambda_i t}$. Note that the eigenvalues of $\tilde{\mathbf{L}}$ is in the range of $[0, 2]$. For $\forall i, j$, if $\lambda_i < \lambda_j$, we have $\frac{g(\lambda_i)}{g(\lambda_j)} = e^{(\lambda_2 - \lambda_1)t} > 1$. Thus $g(\lambda_i) > g(\lambda_j)$. Heat kernel acts as a low-pass filter. As $t$ increases, the ratio $e^{(\lambda_2 - \lambda_1)t}$ also increases, discounting more and more high frequencies. That said, $t$ acts as a modulator between the low frequency and high frequency.

**Different Kernels.** We summary the kernel filters of GCN, SGC, and HKGCN in Table 1. In Figure 2, we use the eigenvalues of $\tilde{L}$ to illustrate the effects of different kernel filters. We can observe that on Cora the absolute values of the filtered eigenvalues by GCN and SGC kernel filters do not decrease monotonically. However, the filtered eigenvalues of heat kernels in HKGCN do monotonically decrease. This is because for $g(\lambda) = (1-\lambda)^k$, it monotonically increases when $\lambda \in [1, 2]$. In other words, the kernel filter of GCN acts as a band-stop filter, attenuating eigenvalues near 1.

Table 1: Kernel Filters.

| Model | Kernel Filter | Propagation |
|---|---|---|
| GCN | $g(\lambda) = 1 - \lambda$ | $\tilde{\mathbf{D}}^{-\frac{1}{2}}\tilde{\mathbf{A}}\tilde{\mathbf{D}}^{-\frac{1}{2}}$ |
| SGC | $g(\lambda) = (1-\lambda)^2$ | $\left(\tilde{\mathbf{D}}^{-\frac{1}{2}}\tilde{\mathbf{A}}\tilde{\mathbf{D}}^{-\frac{1}{2}}\right)^2$ |
| HKGCN $t = 1$ | $g(\lambda) = e^{-\lambda}$ | $e^{-\tilde{\mathbf{L}}}$ |
| HKGCN $t = 2$ | $g(\lambda) = e^{-2\lambda}$ | $e^{-2\tilde{\mathbf{L}}}$ |

**The Influence of High Frequency Spectrum.** The eigenvalues of $\tilde{L}$ and associated eigenvectors satisfy the following relation (Shuman et al., 2016) (Cf Appendix for Proof):

$$\lambda_k = \sum_{(v_i, v_j) \in E} \tilde{\mathbf{A}}_{ij}\left[\frac{1}{\sqrt{\tilde{\mathbf{D}}_{ii}}}\mathbf{u}_k(i) - \frac{1}{\sqrt{\tilde{\mathbf{D}}_{jj}}}\mathbf{u}_k(j)\right]^2 \tag{14}$$

This means that similar to classical Fourier transform, those eigenvectors associated with high $\lambda$ oscillate more rapidly than those associated with low $\lambda$. And we know that $\tilde{\mathbf{L}} = \sum_{i=1}^{n} \lambda_i \mathbf{u}_i \mathbf{u}_i^T$, where $\mathbf{u}_i \mathbf{u}_i^T$ is the projection matrix project to $\mathbf{u}_i$. Since the eigenvector $\mathbf{u}_i$ oscillates more rapidly as the increase of $i$, the projection matrix $\mathbf{u}_i \mathbf{u}_i^T$ will also globally oscillate more rapidly as $i$ increases. Because the filter kernel follows $g(\tilde{\mathbf{L}}) = \sum_{i=1}^{n} g(\lambda_i)\mathbf{u}_i \mathbf{u}_i^T$, the larger $g(\lambda_i)$ is, the greater influence on $g(\tilde{\mathbf{L}})$ will $\mathbf{u}_i \mathbf{u}_i^T$ have. We show above that as $i$ increase, the oscillation of $\mathbf{u}_i \mathbf{u}_i^T$ also increase. Since $\mathbf{Y} = g(\tilde{\mathbf{L}})\mathbf{X}$, an oscillating $g(\tilde{\mathbf{L}})$ will cause oscillating output $\mathbf{Y}$. Therefore, we want the influence of higher $i$'s $\mathbf{u}_i \mathbf{u}_i^T$ as small as possible, i.e. large $i$'s $g(\lambda_i)$ as small as possible. This explains why we need a low-pass filter.

### 3.6 COMPLEXITY ANALYSIS

For the preprocessing step, the time complexity for matrix multiplication between the sparse Laplacian matrix and the feature vector in Chebyshev expansion is $O(d|E|)$, with $|E|$ is the number of edges

and $d$ is the dimension size of input feature. Thus, the time complexity of the preprocessing step is $O(kd|E|)$, with $k$ is the Chebyshev expansion step. For prediction, the time complexity of logistic regression is $O(|V_U|dC)$ with $|V_U|$ is the number of unlabeled nodes, and $C$ is the number of label categories.

The space complexity is $O(|E| + nd)$, where $n$ is the number of nodes.

# 4 EXPERIMENTS

## 4.1 EXPERIMENTAL SETUP

We follow the standard GNN experimental settings to evaluate HKGCN on benchmark datasets for both transductive and inductive tasks. The reproducibility information is detailed in Appendix.

**Datasets.** For transductive learning, we use Cora, Citeseer and Pubmed (Kipf & Welling, 2017; Veličković et al., 2018). For inductive tasks, we use Reddit (Hamilton et al., 2017), ogbn-arxiv (Hu et al., 2020), and a new arXiv dataset collected by ourselves, which contains over one million nodes and 10 million edges. In all inductive tasks, we train models on subgraphs which only contain training nodes and test models on original graphs. We adopt exactly the same data splitting as existing work for Cora, Citeseer, Pubmed, and Reddit.

The ogbn-arxiv dataset is accessed from OGB (`https://ogb.stanford.edu/docs/nodeprop/`). It is the citation network between computer science arXiv papers and the task is to infer arXiv papers' categories, such as cs.LG, cs.SI, and cs.DB. Each paper is given a 128-dimension word embedding based feature vector. The graph is split based on time, that is, papers published until 2017 as training, papers published in 2018 as validation, and papers published in 2019 as

Table 2: Dataset Statistics

|  | Cora | Citeseer | Pubmed | Reddit | arXiv | ogbn-arxiv |
|---|---|---|---|---|---|---|
| #Nodes | 2,708 | 3,327 | 19,717 | 233K | 1.4M | 169K |
| #Edges | 5,429 | 4,732 | 44,338 | 11.6M | 16.4M | 1.2M |
| #Training-Nodes | 140 | 120 | 60 | 152K | 1.1M | 90K |
| #Validation-Nodes | 500 | 500 | 500 | 24K | 121K | 30K |
| #Test-Nodes | 1000 | 1000 | 1000 | 55K | 181K | 49K |
| #Classes | 7 | 6 | 3 | 41 | 175 | 40 |

test. Inspired by ogbn-arxiv, we also constructed a full arXiv paper citation graph from the public MAG (`https://docs.microsoft.com/en-us/academic-services/graph/`). We follow the same feature extraction and data splitting procedures as ogbn-arxiv and generate the arXiv dataset, which will be made publicly available upon publication.

The statistics and splitting information of the six datasets are listed in Table 2.

**Baselines.** For the transductive tasks on Cora, Citeseer, and Pubmed, we use the same baselines used in SGC (Wu et al., 2019), including GCN (Kipf & Welling, 2017), GAT (Veličković et al., 2018), FastGCN (Chen et al., 2018), LanczosNet, AdaLanczosNet (Liao et al., 2019), DGI (Veličković et al., 2019), GIN (Xu et al., 2019b), and SGC (Wu et al., 2019).

For the inductive tasks, we use supervised GraphSage (mean) without sampling (Hamilton et al., 2017), GCN (Kipf & Welling, 2017), ClusterGCN (Chiang et al., 2019), GraphSaint (Zeng et al., 2019), MLP, and SGC (Wu et al., 2019) as baselines.

For the proposed HKGCN, the propagation time $\tilde{t}$ is preset based on the performance on the validation set. On Reddit, we follow SGC (Wu et al., 2019) to train HKGCN and SGC with L-BFGS without regularization as optimizer (Liu & Nocedal, 1989), due to its rapid convergence and good performance. However, this advantage brought by L-BFGS can not be observed in the other datasets, for which we use adam optimizer (Kingma & Ba, 2014), same as the other baselines.

## 4.2 RESULTS

We report the performance of HKGCN and baselines in terms of both effectiveness and efficiency.

**Transductive.** Table 3 reports the results for the transductive node classification tasks on Cora, Citeseer, and Pubmed, as well as the relative running time on Pubmed. As the reference point, it takes 0.81s for training HKGCN on Pubmed. Per community convention (Kipf & Welling, 2017;

Veličković et al., 2018), we take the results of baselines from existing publications (Wu et al., 2019), and report the results of our HKGCN model by averaging over **100** runs. Finally, the efficiency is measured by the training time on a NVIDIA TITAN Xp GPU.

We can observe that 1) HKGCN outperforms SGC in all three datasets with similar training time consumed, and 2) HKGCN can achieve the best prediction results on Citeseer and Pubmed among all methods and comparable results on Cora, while using 2–3 orders of magnitude less time than all baselines except FastGCN and SGC.

In addition, it is easy to notice that comparing to complex GNNs, the performance of HKGCN (and SGC) is quite stable across 100 runs, as it benefits from the the simple and deterministic propagation.

Table 3: Transductive results in terms of test accuracy.

|  | Cora | Citeseer | Pubmed | Pubmed (T) |
|---|---|---|---|---|
| GCN | $81.4 \pm 0.4$ | $70.9 \pm 0.5$ | $79.0 \pm 0.4$ | 25x |
| GAT | $\mathbf{83.3 \pm 0.7}$ | $72.6 \pm 0.6$ | $78.5 \pm 0.3$ | 377x |
| FastGCN | $79.8 \pm 0.3$ | $68.8 \pm 0.6$ | $77.4 \pm 0.3$ | 5x |
| GIN | $77.6 \pm 1.1$ | $66.1 \pm 0.9$ | $77.0 \pm 1.2$ | 81x |
| LanczosNet | $80.2 \pm 3.0$ | $67.3 \pm 0.5$ | $78.3 \pm 0.6$ | 826x |
| AdaLanczosNet | $81.9 \pm 1.9$ | $70.6 \pm 0.8$ | $77.8 \pm 0.7$ | 689x |
| DGI | $82.5 \pm 0.7$ | $71.6 \pm 0.7$ | $78.4 \pm 0.7$ | 236x |
| SGC | $81.0 \pm 0.0$ | $71.9 \pm 0.1$ | $78.9 \pm 0.0$ | 0.9x |
| HKGCN | $81.5 \pm 0.0$ | $\mathbf{72.8 \pm 0.0}$ | $\mathbf{79.9 \pm 0.0}$ | (0.81s) 1x |

**Inductive.** Table 4 summarizes the performance of inductive tasks on large datasets, including both test accuracy and relative running time with HKGCN's as the reference points. Most reported results are averaged over 10 runs, except the supervised GraphSage (mean) and SGC methods' accuracies on Reddit, which are directly taken from the SGC work (Wu et al., 2019).

Table 4: Inductive results in terms of test accuracy (left) and running time (right), averaged over 10 runs.

|  | Reddit | arXiv | ogbn-arxiv | Reddit | arXiv | ogbn-arxiv |
|---|---|---|---|---|---|---|
| GraphSage | 95.0 | 61.0 | 69.4 | 68x | 3.6x | 3.8x |
| GCN | 94.3 | 59.8 | 70.7 | 64x | 2.9x | 2.9x |
| ClusterGCN | 92.6 | 60.7 | 55.9 | 108x | 19.1x | 43x |
| GraphSaint | 92.3 | 59.2 | 56.1 | 138x | 2.4x | 4.0x |
| MLP | 70.0 | 44.1 | 55.5 | 9.3x | 1.75x | 5.2x |
| SGC | 94.9 | 60.1 | 69.6 | 0.5x | 0.93x | 0.95x |
| HKGCN | 95.5 | 60.4 | 70.0 | 1x | 1x | 1x |

The results suggest that 1) the performance HKGCN is consistently better than SGC and comparable to advanced GNN models, such as supervised GraphSage, ClusterGCN, and GraphSaint. Additionally, we notice that HKGCN, GraphSage, and GCN yield the best results on Reddit, arXiv, and ogbn-arxiv, respectively, indicating the lack of universally-best GNN models.

Efficiency wise, HKGCN costs 2.2s, 48.5s, and 5.2s on three datasets, respectively, which are similar to SGC, both of which are clearly more efficient than other GNN models as well as MLP.

**Analysis of $\tilde{t}$.** In HKGCN, $\tilde{t}$ determines how long the feature propagation lasts for. A larger $\tilde{t}$ can amplify global features and a lower one can emphasize local information. From Figure 3, we can observe that $\tilde{t}$ has a similar impact on different datasets, and as the propagation time increases, the model at first benefits from information from its local neighbors. However, as the feature propagation continues (e.g., $\tilde{t} > 9$ or 12), the performance of the model starts to decrease, likely suffering from the over-smoothing issue (Li et al., 2018).

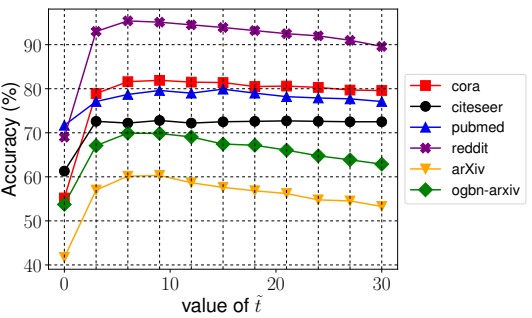

Figure 3: Performances with $\tilde{t}$ varying from 0 to 30.

## 5 CONCLUSION AND DISCUSSION

In this work, we propose to generalize graph convolutional networks (GCNs) and simple graph convolution (SGC) via heat kernel, based on which, we present the HKGCN model. The core idea of HKGCN is to use heat kernel as the feature propagation matrix, rather than the discrete and non-linear feature propagation procedure in GCNs. We theoretically show that the feature propagation of GCNs is equivalent to the finite difference version of heat equation, which leads it to overshoot the convergence point and causes oscillated propagation. Furthermore, we show that

heat kernel in HKGCN acts as a low-pass filter. On the contrary, the filter kernel of GCNs fails to attenuate high-frequency signals, which is also a factor leading to the slow convergence and feature oscillation. While in heat kernel, the cutoff frequency decreases as the increase of the propagation time $t$. Consequently, the HKGCN model could avoid these oscillation issue and propagate features smoothly. Empirical experiments on six datasets suggest that the proposed HKGCN model generates promising results. Effectiveness wise, the linear HKGCN model consistently beats SGC and achieves better or comparable performance than advanced GCN baselines. Efficiency wise, inherited from SGC, HKGCN offers order-of-magnitude faster training than GCNs.

Notwithstanding the interesting results of the present work, there is still much room left for future work. One interesting direction is to learn the propagation time $t$ from data in HKGCN for automatically balancing information between local and global features.

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

# A    REPRODUCIBILITY INFORMATION

## A.1    HYPERPARAMETERS

We introduce the hyperparameters used in the experiments, including weight decay, #epochs, learning rate, $\tilde{t}$, and optimizer, which are also summarized in Table 5. The Chebyshev expansion step $k$ is set to 20.

**Citation Networks.** We train HKGCN for 100 epochs using the Adam optimizer (Kingma & Ba, 2014) with 0.2 as the learning rate, which are the same settings used in SGC (Wu et al., 2019). First, we fix the weight decay to $5 \times 10^{-6}$ and search $\tilde{t}$ from $\{0, 3, 6, 9, 12, 15, 18, 21, 24, 27, 30\}$ based on the performance on the validation set. Then we fix the best performance $\tilde{t}$ and search the weight decay from $\{10^{-6}, 10^{-5}, 10^{-4}\}$ based on the performance on the validation set.

**Reddit dataset.** We follow SGC (Wu et al., 2019) to train HKGCN and SGC with L-BFGS without regularization as the optimizer (Liu & Nocedal, 1989), due to its rapid convergence (2 epochs) and good performance. However, this advantage brought by L-BFGS can not be observed in the other datasets, for which we use the adam optimizer (Kingma & Ba, 2014), same as the other baselines. We search $\tilde{t}$ from all non-negative integers equal or lower than 15, based on the performance on the validation set.

**arXiv and ogbn-arxiv datasets.** We use the Adam optimizer with no weight decay to train HKGCN and SGC, same as other baselines. Using the same treatments in citation networks, we fix the learning rate to 0.2. We first fix 500 epochs and search $\tilde{t}$ from all non-negative integers equal or lower than 15 by using the validation set. On the ogbn-arxiv dataset, the model does not converge after 500 epochs and thus we extend it to 1000.

**Situation for Large $\tilde{t}$.**    If $\tilde{t}$ is greater than 10, the floating point precision error would become an issue in calculating the Bessel function. Therefore, in practice, we convert $e^{(-\bar{L}\tilde{t})}$ into $e^{(-\bar{L}\frac{\tilde{t}}{3})}e^{(-\bar{L}\frac{\tilde{t}}{3})}e^{(-\bar{L}\frac{\tilde{t}}{3})}$.

**Situation for Very Large Graphs.** Though the Chebyshev expansion is very close to the minimax polynomial on the range of $[-1, 1]$ and thus makes it converge very fast, the recurrence relation in Eq. 13 is second-order. In other words, the calculation of $T_{i+1}(\mathbf{x})$ needs to store $T_i(\mathbf{x})$, $T_{i-1}(\mathbf{x})$ and $\tilde{\mathbf{L}}$, making it require more GPU memory than the first-order recurrence relation expansion, such as Taylor expansion. Therefore, instead of using the Chebyshev expansion, we leverage Eq. 7 as $\mathbf{X}^{(t+\Delta t)} = (\mathbf{I} - \Delta t\tilde{\mathbf{L}})\mathbf{X}^{(t)}$ for the calculation. In this way, we only need to store $\mathbf{X}^t$ and $\tilde{\mathbf{L}}$, which requires much less memory space.

**Training Time** . We tested all experiments on NVIDIA TITAN Xp GPU with 12 GB memory. The training time of HKGCN is the sum of 1) the feature pre-processing time and 2) logistic regression training time. So we can see the largest training time difference between SGC and HKGCN occurs on the Reddit dataset. This is because it only takes two epochs to converge on the Reddit dataset by using L-BFGS, making the logistic regression step very fast. Thus the pre-processing step dominates the training time for HKGCN and SGC on Reddit.

Table 5: Hyperparameters

|  | Cora | Citeseer | Pubmed | Reddit | arXiv | ogbn-arxiv |
|---|---|---|---|---|---|---|
| $\tilde{t}$ | 12 | 9 | 15 | 6 | 6 | 8 |
| weight decay | $10^{-5}$ | $10^{-4}$ | $10^{-5}$ | n/a | 0 | 0 |
| #epochs | 100 | 100 | 100 | 2 | 500 | 1000 |
| learning rate | 0.2 | 0.2 | 0.2 | 1 | 0.2 | 0.2 |
| optimizer | adam | adam | adam | l-bfgs | adam | adam |

## A.2    BASELINE METHODS

Per community convention (Kipf & Welling, 2017; Veličković et al., 2018), the results of baselines in the three citation networks (Cora, Citeseer, and Pubmed) are taken from existing publications (Wu

et al., 2019). The hyperparameters settings of inductive tasks on three large datasets (Reddit, arXiv, and ogbn-arxiv) are introduced as follow.

**SGC.** To make a fair comparison, we select $K$ in SGC based on the performance on the validation set, which has similar meaning as $\tilde{t}$ in HKGCN.

**GCN.** The learning rate is set to 0.01 on all datasets. The weight decay is 0. The number of layers is 3 on Reddit and ogbn-arxiv, and 2 on arXiv because of the GPU memory limit. The hidden layer size is 256 on Reddit and ogbn-arxiv, and 128 on arXiv also due to memory limit. The number of epochs is 500.

**GraphSage.** Results on the Reddit dataset are directly taken from the SGC paper (Wu et al., 2019). The other parameter settings are the same as GCN above. We use the mean-based aggregator as it provides the best performance.

**MLP.** The number of layers is 3 and the hidden layer size is 256, and Batch Norm is not used.

**ClusterGCN.** On the Reddit dataset, the number of partitions is 1500, the batch size is 20, the hidden layer size is 128, which are the same parameter settings as the original SGC paper (Chiang et al., 2019). On the arXiv dataset, the number of partitions is 15000, the batch size is 32, the hidden layer size is 256. The number of epochs of these two datasets is 50. On the ogbn-arxiv dataset, the number of partitions is 15 (as a large number would cause an unknown bug in the PyTorch-Geometric library), the batch size is 32, the hidden layer size is 256. We set the epoch number to 200 on the ogbn-arxiv dataset because it does not converge after 50 epochs.

**GraphSaint.** On the Reddit dataset, the number of layers is 4, the hidden layer size is 128, the dropout rate is 0.2, the same parameter settings as used in SGC (Zeng et al., 2019). On the arXiv dataset, the number of layers is 3, the hidden layer size is 256, the dropout rate is 0.5. On the ogbn-arxiv dataset, we use the same parameter setting as the Reddit dataset, because both graphs are on the same scale.

## B  ALGORITHM OF PRESETTING $t$

---

**Algorithm 1** Preset $t$ in HKGCN

---

**Input:** input features $\mathbf{X}$; feature propagation duration time $\tilde{t}$; Chebyshev expansion step $k$; scaling augmented normalized Laplacian $\bar{\mathbf{L}} = \tilde{\mathbf{L}}/2$; $B_i(t)$ is the modified Bessel function.
**Output:** pre-processed features $\mathbf{X}^{(t)}$;
    $T_0(\mathbf{x}) \leftarrow \mathbf{X}$;
    $\mathbf{X}^{(t)} \leftarrow B_0(\tilde{t})T_0(\mathbf{x})$;
    $T_1(\mathbf{x}) \leftarrow \bar{\mathbf{L}}\mathbf{X}$;
    $\mathbf{X}^{(t)} \leftarrow \mathbf{X}^{(t)} - 2B_1(\tilde{t})T_1(\mathbf{x})$;
    **for** $i = 2...k$ **do**
        $T_i(\mathbf{x}) \leftarrow 2\bar{\mathbf{L}}T_{i-1}(\mathbf{x}) - T_{i-2}(\mathbf{x})$;
        **if** $i$ is odd **then**
            $\mathbf{X}^{(t)} \leftarrow \mathbf{X}^{(t)} - 2B_i(\tilde{t})T_i(\mathbf{x})$;
        **else**
            $\mathbf{X}^{(t)} \leftarrow \mathbf{X}^{(t)} + 2B_i(\tilde{t})T_i(\mathbf{x})$;
        **end if**
    **end for**

---

# C  PROOF OF EQ. 14

$$
\begin{aligned}
\lambda_k &= \mathbf{u}_k^T \tilde{\mathbf{L}} \mathbf{u}_k \\
&= \mathbf{u}_k^T (\mathbf{I} - \tilde{\mathbf{D}}^{-\frac{1}{2}} \tilde{\mathbf{A}} \tilde{\mathbf{D}}^{-\frac{1}{2}}) \mathbf{u}_k \\
&= \sum_{i \in \{1,\ldots,n\}} \mathbf{u}_k(i)^2 - \sum_{(v_i,v_j) \in E} \frac{2\tilde{\mathbf{A}}_{ij}}{\sqrt{\tilde{\mathbf{D}}_{ii}\tilde{\mathbf{D}}_{jj}}} \mathbf{u}_k(i)\mathbf{u}_k(j) \\
&= \sum_{(v_i,v_j) \in E} \tilde{\mathbf{A}}_{ij} \left[ \frac{1}{\sqrt{\tilde{\mathbf{D}}_{ii}}} \mathbf{u}_k(i) - \frac{1}{\sqrt{\tilde{\mathbf{D}}_{jj}}} \mathbf{u}_k(j) \right]^2
\end{aligned}
$$

