# OpenReview forum: "Generalizing Graph Convolutional Networks via Heat Kernel"
_ICLR.cc/2021/Conference — Reject_

### Official Review · AnonReviewer4 · 2020-10-28
**Heat-kernel GCN**

**Rating:** 6
**Confidence:** 5

**Review:**

This submission proposes to use heat kernel as the propagation matrix in graph convolutional networks. The authors show that heat kernels can induce more smooth propagation behavior than the commonly used discrete propagation. The submission also designs an efficient method to calculate the heat kernel (using Chebyshev expansion).

Strength:
- The proposed method is well motivated and heat kernel is (relatively) well-understood.
- The writing is clear and easy-to-understand
- The proposed method demonstrates improved performance on node classification datasets while inheriting the efficiency of simple graph convolution
- The submission collects a larger scale arxiv graph dataset, which can be useful for the community. Will the authors release this dataset in the future?
- This submission has done a detailed and thorough evaluation of the proposed method. I appreciate the effort to analyze $\tilde{t}$, which should be helpful for practitioners.

Weakness:
- SGC is known to be ineffective for graph classification tasks. Does the proposed model also inherits this downside?
- Table 3 lacks comparisons to more recent graph neural networks like Graph Markov Neural Networks. However, I don't think this is critical given the efficiency and strong performance of the proposed method on larger graph datasets.


=== UPDATED ===

Given that the reviewers have not reached a consensus, I want to add more discussion to my review to facilitate the AC to make the decision. I would also include a few more quick TODOs for the authors and hope they can help add evidence to my argument.

1. This submission proposes to replace the propagation in GNNs with heat kernel. The main motivation for this method is that the laplacian filters tend to oscillate, as illustrated by Figure 1. The heat-kernel provides a continuous convergence process and intuitively may address the oscillation process. Importantly, I believe the motivation of this method is *not* to prevent oversmoothing but to prevent over-oscillation.

I believe this intuition is sound but encourage the authors to do more to validate this hypothesis. I appreciate the ablation study in Figure 6. As one more analysis, I suggest the authors to add the performance curves of SGC to Figure 6 (under the same setting). If this theoretical intuition is valid, we should expect HKGCN and SGC to behave more differently when the propagation degree is low (within 2-10). My understanding is that both HKGCN and SGC are efficient so this experiment shouldn't take long. Ideally the authors can update this result, at least on 1-2 datasets, within the discussion period.

2. I am also very impressed by the efficiency of HKGCN. This submission has experimented with, to my knowledge, the largest publicly available graph dataset (arXiv), which contains more than one million nodes. According to the authors, HKGCN can be trained for this arXiv dataset in 48.5s, which is impressive.

Note that here the heat diffusion matrix does not need to be computed/stored explicitly. Following the setup in SGC, HKGCN only needs to compute the propagated features in a preprocessing step.

I suggest the authors to give more concrete numbers to illustrate the efficiency of this method. For this arXiv dataset, what kind of hardware is required? How much actual RAM did you use to compute the preprocessing step?

3. After reading other reviews, I now realize that this submission is not the first to introduce heat kernels into GCNs. Among the papers pointed out by other reviewers, I find [1] to be most relevant in that they also proposed the usage of heat kernels. Can the authors also clarify the difference between this submission and [2]?

Based on this novelty concern, I have lowered my review score to 6.

[1] Xu et al, graph wavelet neural network, ICLR 2019
[2] Xu et al, Graph Convolutional Networks using Heat Kernel for Semi-supervised Learning, IJCAI 2019.

---

> ### Author Response · Authors · 2020-11-20
> **Response to AnonReviewer4**
>
> Thanks for your positive feedback!
>
>
> **Reviewer Comment: SGC is known to be ineffective for graph classification tasks. Does the proposed model also inherits this downside?**
>
> Our Response: For graph classification tasks, we usually need a graph readout layer to exact features from the whole graph. The ability to distinguish graphs mainly depends on this readout layer, instead of graph convolution. In this paper, we mainly target at node classification, under the setting of mean aggregator in GCN and SGC. Readout layer could be integrated to solve this problem.
>
>
> **Reviewer Comment: Table 3 lacks comparisons to more recent graph neural networks like Graph Markov Neural Networks.**
>
> Our Response: Thank you for giving this suggestion. More comparisons could be helpful.

---

### Official Review · AnonReviewer3 · 2020-10-29
**Simple Graph Convolution (SGC) + Heat Diffusion**

**Rating:** 5
**Confidence:** 4

**Review:**

This submission introduced a new graph convolutional operator based on heat diffusion, named heat kernel GCN (HKGCN). First, continuous-time heat diffusion on graphs is reviewed, where the solution is given by the heat equation (6). Then, the authors showed that classical GCN can be approximated in the same formulation through discretization.

The proposed method HKGCN is similar to the simplified GCN (SGC) by Wu et al. The learning procedure is quite straightforward: first, a heat diffusion is performed on the input features for a pre-specified time $t$; then, logic regression is performed on the diffused features to train the classifier. The SGC model performs several times of neighborhood averaging (as in vanilla GCN) based on a polynomial spectral filter of order one; while HKGCN performs heat diffusion with higher-order polynomial terms.

My main criticism is that applying heat diffusion on graph convolution is not new and the relationship with previous works is not clearly stated. See the cited (Xu et al 2019a) or not mentioned (Klicpera 2019) (Xu et al 2019c), where similar formulations and ideas already appeared. The main novelty of the proposed HKGCN, therefore, is on a combination of heat kernel and the SGC approach. This combination is not non-trivial enough and may not be significant enough to be published in ICLR.

The HKGCN method is motivated by the oscillation problem of GCN. How does the heat kernel help avoid oscillation? After the introduction of the heat equation, there should be some theoretical statements to provide a solution to this problem and to correspond to this motivation. Ideally, to show how HKGCN is different with the GCN approach. Instead, the oscillation problem is mainly solved by numerical simulation on the toy example and informal arguments.

As another novelty, the authors revealed that GCN can be approximated using heat diffusion under the same formulation. Again the connection although interesting is not a major contribution.

Empirically, the authors tested the HKGCN method on commonly used citation datasets and an OGB graph of arxiv articles, on both transitive and inductive learning tasks.

The huge speed improvement is mainly due to the same trick as SGC is used in HKGCN: there is no activation between the convolution layers which allows a pre-computation step followed by an extremely simplified learning step (logistic regression). This improvement is due to SGC and is expected.

By looking at the accuracy scores, the main comparison is HKGCN vs SGC because of their similarities. It is not convincing that using heat diffusion (HKGCN) instead of graph convolution (SGC) can bring a notable performance improvement. In most of the time, the improvement is quite marginal.

Overall, this technical novelty and empirical significance are limited. There are not theoretical statements in this paper and I am evaluating it as an algorithmic contribution. I am recommending a weak rejection.

More comments:

The title is too broad. Please be more specific.

Toy example in the introduction: As this toy is mentioned again in later text, please explain in more detail the computation of GCN vs HKGCN.

As you started from GNN, it is good to cite some original GNN paper (Gori et al. 2005; Scarselli et al 09)

eq.(9) n_{iter} is hard to read

in this template, most of the citations should use \citep instead of \cite

References:

(Klicpera 2019)
"Diffusion Improves Graph Learning", Klicpera et al. 2019

(Xu et al 2019c)
"graph wavelet neural network" Xu et al. 2019

---
After rebuttal:

Thank you for the revision and the clarifications.

"no one has made a clear connection between GCN and the heat kernel."
For example, in (Klicpera 2019) section 2, the heat kernel is discussed as a special case.

"We don’t simply combine SGC and heat kernel."
Clearly, the only difference between the proposed method in section 3.4 and SGC is that the authors used heat diffusion as the spectral filter matrix, while SGC used a polynomial filter (the K'th power of the normalized adjacency matrix).

Furthermore, the other reviewers raised similar works such as graph ODE, which further reduces the novelty of this work.

---

> ### Author Response · Authors · 2020-11-20
> **Response to AnonReviewer3**
>
> Thanks for providing very valuable feedback regarding our submission and indicating specific points of uncertainty. Please see our response below:
>
>
> **Reviewer Comment: The relationship with previous works is not clearly stated.**
>
> Our Response: The previous works did not make a clear connection between GCN and the heat kernel, which is what HKGCN does. We will discuss the listed papers in details as follows.
>
> (Xu et al 2019c) proposed using graph wavelet transform as bases. It isn’t quite relevant to our model which is focused on improving graph feature propagation. As discussed in the paper, (Xu et al 2019a) first introduced heat kernel into GCNs. (Klicpera 2019) generalized graph diffusion and used sparsification to decrease graph density. But we would argue that no one has made a clear connection between GCN and the heat kernel. We don’t simply combine SGC and heat kernel. We use SGC as baseline because SGC removed nonlinear activation function, so it is a great reference model to show the ability of the heat kernel to generalize multi-layer GCN.
>
>
> There are three main aspects which haven't been raised by previous works. First, we introduced an appropriate inductive bias that the propagation of features in graphs also follows Newton’s law of cooling, which means the propagation rate on each edge is proportional to the difference between the features of nodes it connects. Second, we theoretically proved that the propagation matrix in GCN or SGC can be seen as a discrete version of the heat kernel. Third, although (Xu et al 2019a) (Klicpera 2019) also proposed that heat kernel is a low-pass filter. We go further to identify the reason why low-pass filter prevents oscillation, compared with band-stop filter(GCN).
>
>
> **Reviewer Comment: How does the heat kernel help avoid oscillation?**
>
> Our Response: The reason for how the heat kernel helps avoid oscillations can be summarized into two aspects, vertex domain and spectral domain.
>
> First, in vertex domain, heat kernel can be seen as GCN with step size of infinitely small. This helps avoid overshooting the convergence point, which is caused by too large step size. Just like we don’t want to use a large learning rate in deep learning.
>
>
> Second, in our spectral analysis section, we prove that high frequency spectrum will cause globally oscillating, which will lead to oscillations in features. So the low-pass filter (heat kernel) performs better in avoiding oscillations than the band-stop filter(GCN).
>
>
> **Reviewer Comment: The title is too broad. Please be more specific.**
>
> Our Response: We will change it to “Generalizing Graph Convolutional Networks via Heat Kernel” in our updated version.
>
>
> **Reviewer Comment: Toy example in the introduction: As this toy is mentioned again in later text, please explain in more detail the computation of GCN vs HKGCN.**
>
> Our Response: The reason why we use this toy example is that we want to give an intuitive feeling about the ability of HKGCN to prevent oscillation. We also theoretically proved that in bothe vertex and spectral domain.
>
>
> **Reviewer Comment: As you started from GNN, it is good to cite some original GNN paper (Gori et al. 2005; Scarselli et al 09)**
>
> Our Response: Thank you. We will discuss them in our updated version.
>
>
> **Reviewer Comment: eq.(9) n_{iter} is hard to read. in this template, most of the citations should use \citep instead of \cite**
>
> Our Response: Thank you. We have fixed that in our updated version of paper.

---

### Official Review · AnonReviewer2 · 2020-11-01
**Interesting work, but novelty is limited, and very relevant references are not cited and compared**

**Rating:** 5
**Confidence:** 5

**Review:**

This paper studies semi-supervised node classification in graph data. One powerful approach to the task is graph convolutional networks, which use discrete layers to perform information propagation. The paper generalizes GCNs into a continuous model via heat kernel, where the proposed model uses continuous layers for information propagation. The authors conduct both theoretical and empirical analysis of the proposed model. Experiments on several standard datasets show promising results. Overall, the paper studies an important problem in graph machine learning, and proposes a principled approach, which combines graph neural networks with heat kernels, and gives a new way of analyzing existing graph neural networks.
However, the paper also has several weaknesses:
1. The novelty is limited.
Although the idea of developing continuous propagation layers is interesting, the idea has been explored by many recent works. For example, [1,2,3] use a graph neural network to define an ODE, which leads to a continuous feature propagation layer for node classification. [4] uses a linear ODE for feature propagation, which is very similar to the method proposed here, and the only difference is that the ODE in [4] also incorporates some constant term besides -LX. The authors should explain and clarify the difference between this work and existing works.

2. The results are worse than SOTA methods.
In experiments, the authors conduct experiments in many standard datasets (e.g., Cora, Citeseer, Pubmed), and the proposed method shows promising results. However, the compared methods used in experiments are not competitive enough. The strongest baseline methods in Table 3 are GCN and GAT, and in Table 4 they are GraphSage and GCN. All these methods are proposed before 2017, and recently there are many more competitive methods proposed. To make the results more convincing, it is helpful to compare against some recent graph neural networks for node classification.
Besides, I also have some questions regarding the model detail:
1. About t in equation (6).
In Equation (6), the analytical form of X^{(t)} is given by X^{(t)}=e^{-Lt}X, where t can have a high impact on the results. If t is very small, then e^{-Lt} becomes an identity matrix, and hence H_t will be very close to X. If t is very large, then e^{-Lt} becomes an matrix whose elements are all close to 0, and thus all the rows in H_t will be almost the same, yielding an over-smoothing problem. In practice, what would be a proper value of t? Moreover, if we look at Figure (6), even when t is very large (e.g., t>20), the accuracy is still very high especially on Cora, which indicates that the model does not suffer from over-smoothing in practice. But if we check the analytical form X^{(t)}=e^{-Lt}X, when t is large, all the rows in H_t become very similar, which may lead to over-smoothing and a low accuracy. I wonder how does the proposed model manage to avoid over-smoothing in practice? Could the authors elaborate on that?
2. About feature dimensionality.
In the propose method, the hidden matrix H_t has the same size as the feature matrix X. If the feature dimensionality of a dataset is very high, which is quite common in practice, then computing H^{(t)} can entail high cost. Is there a way to deal with the potential problem?
3. About the time complexity.
In Section 3.6, the authors mention that the time complexity of data processing is O(k|E|). What is k here?
References:
[1] Poli, Michael, et al. "Graph neural ordinary differential equations." arXiv preprint arXiv:1911.07532 (2019).
[2] Deng, Zhiwei, et al. "Continuous graph flow for flexible density estimation." arXiv preprint arXiv:1908.02436 (2019).
[3] Zhuang, Juntang, et al. "Ordinary differential equations on graph networks." (2019).
[4] Xhonneux, Louis-Pascal AC, Meng Qu, and Jian Tang. "Continuous Graph Neural Networks." arXiv preprint arXiv:1912.00967 (2019).

---

> ### Author Response · Authors · 2020-11-20
> **Response to AnonReviewer2 Part1**
>
> Thank you for your detailed and insightful comments! Please see our response below.
>
> **Reviewer Comment: Not novel, what is the differences between this work and [1,2,3,4]?**
>
> Our Response: [1,2,3] are different from HKGCN in both motivation and implementation. The formulas in [4] looks similar but different in meaning. We will discuss in details as follows:
>
> [2] use continuous flow on a generative model to solve generation tasks, but HKGCN aims to classification. [1, 3] use the Neural ODE framework and parametrize the derivative function using a 2 or 3 layer GNN directly. [4] improved ODE by developing a continuous message-passing layer.
>
> From a small beginning we can see how things will develop. We could find all their layers(nodes) have a self-loop. For example, in [1] equation (2), they turned GCN into the residual version, same as equation (3) in [3]. In equation (4) of [4], each node also needs to remember its original node features $H_0$. They do so because they couldn’t make their models converge to a stable fixed point without adding a residual connection. This also shows why $t$ in [4] has quite a different meaning from ours $t$. They make their $t \to \infty$ to enable their networks to have an infinite number of ‘layers’. But our $t$ is a balance between local feature and global feature, so it shouldn’t be too large nor too small. We didn’t want to make our features converge to a stable point. We just want to use $t$ to control the propagation rate(time) of the nodes.
>
>
> A reviewer of [3] said it would be better if the authors could build a more detailed relationship between the neural ODEs framework and underlying GNN. He was also curious about how the topological information of graphs affects the graph ODEs via spectral-type graph convolution operations and what is the relationship to the oversmoothing phenomena. We think this is exactly what we do. We showed the relationship between heat kernel and gcn. We also analyzed from the spectral perspective and gave reasons why the low-pass filter will prevent oscillation.
>
> We will add a more detailed discussion in the camera-ready version.
>
> **Reviewer Comment: The results are worse than SOTA methods.**
>
> Our Response: Our main contribution is to generalize GCNs into a continuous and linear propagation model with theoretical analyses. We are not focusing on getting the highest SOTA performance. We improve the propagation matrix in feature message passing. We fixed everything else in SGC, except replacing the feature propagation matrix with the heat kernel. Our theoretical analyses part is also focused on what makes the heat kernel a better feature propagation matrix. Our model hyperparameters are listed in the appendix. Please feel free to ask if you have any questions about model detail.
>
>
> **Reviewer Comment: Why large t won’t cause low accuracy and how does the model avoid over-smoothing in practice?**
>
> Our Response: Thank you for pointing out this very interesting phenomenon. I think the reason behind this is still the continuous model. $H_t$ did tend to each element equals to $1/n$ as $t$ grows. And this makes the variation of nodes feature $var(X^{(t)})$ tends to zero. But because of the continuous model, the relationship between features of nodes are preserved. It’s like we shift all nodes’ features with the same rate toward the average feature of the whole graph as $t$ grows. This makes our linear logistic regression classifier still have the capability to distinguish different nodes. Our toy example in Figure 1 provides a more direct feeling, in which the node with larger features is always the same node in our model and GCN on the contrary.
>
>
> **Reviewer Comment: How to deal with computation cost when feature dimension is high?**
>
> Our Response: The size of matrix $H_t$ is $n*n$, n is the number of nodes in the graph. The way to calculate $H_t * X$ is by using Chebyshev expansion to convert $H_t$ into a polynomial of $A$, which is the adjacency matrix. Because $AX$ is a sparse matrix multiplication, the time complexity is proportional to the dimension of input feature, same as GCN. So computing $H_t X$ is proportional to the dimension of input feature, which won’t cost a lot of time. Thanks for pointing out we missed calculating the dimension of input features in time complexity of the preprocessing step. We have changed it in our updated version of paper.

---

> > ### Author Response · Authors · 2020-11-20
> > **Response to AnonReviewer2 Part2**
> >
> > **Reviewer Comment: In Section 3.6, the authors mention that the time complexity of data processing is O(k|E|). What is k here?**
> >
> > Our Response: k is the Chebyshev expansion step in Equation 13. So as the Chebyshev expansion step increases, the time complexity will also increase linearly. Thanks for this advice. We emphasized this in our updated version of paper.
> >
> > [1] Poli, Michael, et al. "Graph neural ordinary differential equations." arXiv preprint arXiv:1911.07532 (2019). [2] Deng, Zhiwei, et al. "Continuous graph flow for flexible density estimation." arXiv preprint arXiv:1908.02436 (2019). [3] Zhuang, Juntang, et al. "Ordinary differential equations on graph networks." (2019). [4] Xhonneux, Louis-Pascal AC, Meng Qu, and Jian Tang. "Continuous Graph Neural Networks." arXiv preprint arXiv:1912.00967 (2019).

---

### Official Review · AnonReviewer1 · 2020-11-01
**Establishing link between heat kernels and linear GCN for more expressive graph representations**

**Rating:** 6
**Confidence:** 4

**Review:**

The authors  shed light on  linear GCNs models and propose  a new design which aim at generalizing  GCNs to continuous and  and a linear propagation model inspired by Newton's law.
It is based on the hypothesis that features propagation across nodes in a given graph follows the same process.
 To do so, the authors establish a link with heat kernels and formulate the problem as heat kernel learning within linear GCN model so that the network at the feature propagation step takes into consideration  multi-hop neighboring systems to refine the features of a given central node.

In this paper, the authors find out that features propagation based on heat kernel allows to control the oscillation between low and high frequencies. Controlling the appropriate level of granularities is quite a challenging task in deep learning mainly, as the convolutional filters are biased toward low frequencies.
However,  important invariants and informative information for classification are within the chaos of high frequencies. In order to control that, the authors explain that  GCN based heat kernels can act as a low-pass filter cutoff. This combination of GCN and heat kernels are empirically validated considering node and graph classification tasks. The settings are clear and the comparison with related works is convincing.

One of the strong points of the paper is its capacity to provide comparable results state-of-the-art  with a reasonable complexity in space, with the advantage of being more simple and interpretable compared to existing (related) methods. Moreover a theoretical analysis from a spectral standpoint is introduced clearly.  It consists at setting link between Linear GCN and heat kernels, as well as with finite difference methods.
However, it’s not clear how the proposed GCN tackles the problem of over-smoothness and graph isomorphism. They are among the most challenging problems in graph learning. From that, l derive two questions :

1- To what extent GCN based on heat kernel formulation is able to mitigate over-smoothness ?

2- Is the proposed heat kernel function injective so that it is able to distinguish two non-isomorphic graphs ?

One possible weakness of the proposed design, is that it can be applicable only on graphs with fixed topology and size as it is the case for all spectral GCN (filters are not transferable across graphs since they are basis dependent). However, in real world problems  graphs are irregular.

The overall approach is  original, well placed in the litterature and the paper is well written. The authors conduct both theoretical and practical studies to show that this research direction could be important to improve existing GCN models. For that reason, l propose to accept the paper.

---

> ### Author Response · Authors · 2020-11-20
> **Response to AnonReviewer1**
>
> Thank you for the detailed comments. We provide detailed answers to each question as follows:
>
>
> **Reviewer Comment: To what extent GCN based on heat kernel formulation is able to mitigate over-smoothness?**
>
> Our Response: The experiment in Figure 3 showed our model is able to mitigate over-smoothness. Our features are still distinguishable and the model performs well even when \Tilde{t} = 30 (\Tilde{t}=30 is equivalent to a 15-layers GCN).
> In [1], they said that the reason which causes over smoothing is that “repeated applying Laplacian smoothing may mix the features of vertices from different clusters and make them indistinguishable”. However, we improved Laplacian from discrete multiplications to continuous exponential. It’s like we shift all nodes’ features with the same rate toward the average feature of the whole graph as $t$ grows. This makes our linear logistic regression classifier still have the capability to distinguish different nodes. Our example in Figure 1 provides a more direct feeling, in which the node with larger features is always the same node in our model and GCN on the contrary. This shows how our model prevents oscillation and makes our features still distinguishable as \Tilde{t} increases.
>
> [1] Li, Qimai, Zhichao Han, and Xiao-Ming Wu. "Deeper insights into graph convolutional networks for semi-supervised learning." arXiv preprint arXiv:1801.07606 (2018).
>
>
> **Reviewer Comment: Is the proposed heat kernel function injective so that it is able to distinguish two non-isomorphic graphs?**
>
> Our Response: To distinguish two non-isomorphic graphs, or more generally the "graph classification" tasks, we usually need a graph readout layer to exact features from the whole graph. The ability to distinguish graphs mainly depends on this readout layer, instead of graph convolution. In this paper, we mainly target at node classification, under the setting of mean aggregator in GCN and SGC. Standard techniques for graph isomorphism, e.g. WL test, can be integrated into our method during the readout period to solve this task.
>
>
> **Reviewer Comment: It can be applicable only on graphs with fixed topology and size as it is the case for all spectral GCN (filters are not transferable across graphs since they are basis dependent). However, in real world problems graphs are irregular.**
>
> Our Response: The inductive results in Table 4 show that HKGCN generalize its parameter from a smaller subgraph which only contains training nodes to the whole graph, exhibiting its flexibility under **subgraph-level inductive learning**.
> However, just as you say, the graph-level inductive learning is a problem that nearly all GNNs are faced, if your "irregular" means that graphs will change greatly over the time.

---

### Decision · Program_Chairs · 2021-01-07
**Final Decision**

**Decision:**

Reject

**Comment:**

This paper has been evaluated by four reviewers who overall hesitated between borderline reject/accept. In general, as Rev. 4 points out, this paper appears to cope with over-oscillation rather than over-smoothing aspect of GCN modeling (something worth clarifying). Rev. 3 also rightly points out that the connection between the heat kernel and GCN in fact was established in previous works. Also, the connection between SGC (polynomial filter) and  the heat diffusion (the spectral filter matrix) is hard to overlook (the impression that this work builds heavily on SGC). Therefore, while AC sympathizes with the idea, it is also difficult to overlook the incremental nature of the paper and therefore the paper cannot be accepted in its current form.